# The Inhibition of Serine Proteases by Serpins Is Augmented by Negatively Charged Heparin: A Concise Review of Some Clinically Relevant Interactions

**DOI:** 10.3390/ijms25031804

**Published:** 2024-02-02

**Authors:** Edward D. Chan, Paul T. King, Xiyuan Bai, Allen M. Schoffstall, Robert A. Sandhaus, Ashley M. Buckle

**Affiliations:** 1Department of Medicine, Rocky Mountain Regional Veterans Affairs Medical Center, Aurora, CO 80045, USA; 2Department of Academic Affairs, National Jewish Health, Denver, CO 80206, USA; 3Division of Pulmonary Sciences and Critical Care Medicine, University of Colorado School of Medicine, Aurora, CO 80045, USA; 4Medicine Monash Health, Monash University, Clayton, VIC 3800, Australia; 5Department of Chemistry and Biochemistry, University of Colorado, Colorado Springs, CO 80918, USA; 6Department of Medicine, National Jewish Health, Denver, CO 80206, USA; 7Department of Biochemistry and Molecular Biology, Biomedicine Discovery Institute, Monash University, Clayton, VIC 3800, Australia; abuckle@replay.bio; 8Replay, San Diego, CA 92121, USA

**Keywords:** serine protease, serpin, TMPRSS2, heparin, glycosaminoglycans, C1-esterase inhibitor, antithrombin III, SARS-CoV-2

## Abstract

Serine proteases are members of a large family of hydrolytic enzymes in which a particular serine residue in the active site performs an essential role as a nucleophile, which is required for their proteolytic cleavage function. The array of functions performed by serine proteases is vast and includes, among others, the following: *(i)* the ability to fight infections; *(ii)* the activation of blood coagulation or blood clot lysis systems; *(iii)* the activation of digestive enzymes; and *(iv)* reproduction. Serine protease activity is highly regulated by multiple families of protease inhibitors, known collectively as the SERine Protease INhibitor (SERPIN). The serpins use a conformational change mechanism to inhibit proteases in an irreversible way. The unusual conformational change required for serpin function provides an elegant opportunity for allosteric regulation by the binding of cofactors, of which the most well-studied is heparin. The goal of this review is to discuss some of the clinically relevant serine protease–serpin interactions that may be enhanced by heparin or other negatively charged polysaccharides. The paired serine protease–serpin in the framework of heparin that we review includes the following: thrombin–antithrombin III, plasmin–anti-plasmin, C1 esterase/kallikrein–C1 esterase inhibitor, and furin/TMPRSS2 (serine protease Transmembrane Protease 2)–alpha-1-antitrypsin, with the latter in the context of COVID-19 and prostate cancer.

## 1. Introduction to Serine Proteases and Serine Protease Inhibitors

Serine proteases are members of a large family of hydrolytic enzymes that cleave peptide bonds in two steps, in which a particular serine residue in the active site of the serine protease performs an essential role as a nucleophile, which is required for their proteolytic cleavage function [1,2]. Firstly, a lone pair of electrons of the nucleophilic hydroxyl group of serine in the serine protease intermolecularly attacks the electrophilic carbonyl carbon of the scissionable peptide bond of the peptide–protein substrate, freeing the C-terminal peptide fragment. Secondly, the electrons of the nucleophilic water molecule attack the same carbon of the original peptide, regenerating the serine moiety on the serine protease and freeing the N-terminal peptide fragment. During this process, a histidine group of the serine protease facilitates the serine attack by accepting the freed hydrogen ion when the electrons from the serine hydroxyl group attack the carbonyl carbon of the peptide bond and by donating a hydrogen ion to the nitrogen of the peptide leaving group.

The array of salubrious functions performed by serine proteases is vast and includes the following: *(i)* the ability to fight infections via direct antimicrobial effects (attacking bacterial membranes, inactivating bacterial virulence factors, and helping to form the neutrophil extracellular traps (NETs)) or indirectly by the activation of the complement system and by generation of antimicrobial peptides [3]; *(ii)* the activation of the blood coagulation or blood clot lysis systems; *(iii)* the activation of digestive enzymes; *(iv)* reproduction, etc. In contrast to these beneficial effects for the host, some cell surface serine proteases cleave viral envelope proteins, a process that is necessary before viruses fuse with the cell membrane and gain intracellular entry.

Serine protease activity is highly regulated by multiple families of protease inhibitors. In contrast to the competitive or ‘lock-and-key’ inhibitors such as Kunitz-type inhibitors [4], members of the SERine Protease INhibitor (SERPIN) superfamily use a conformational change mechanism to inhibit proteases in an irreversible way. The native serpin fold is composed of nine α-helices, three β-sheets and an extended Reactive Center Loop (RCL), representing a kinetically trapped metastable conformation that uses the extended RCL as ‘bait’ for the serine proteases. The serpin’s inhibition mechanism is akin to a “molecular mousetrap”, marked by significant structural changes during its molecular transformation [5]. This process begins when the protease binds to the serpin’s RCL, creating a reversible Michaelis–Menten complex. As the protease’s serine-195 attacks the RCL’s P1 residue, the interaction shifts from a reversible to a stable covalent bond, breaking the P1-P1′ peptide bond and triggering the release of the RCL’s N-terminal region, akin to a spring, with the protease still attached. This leads to a critical juncture where the serpin can either continue inhibiting or become a substrate for the protease. The outcome hinges on how quickly the RCL integrates between strands 3 and 5 of the A β-sheet. If this integration outpaces the protease’s de-acylation (which is the completion of its catalytic cycle), the serpin remains an inhibitor. In this scenario, the still-bound protease is moved a significant distance (71 Å) across the serpin, distorting its active site. This movement pulls the catalytic serine away from its partner, histidine-57, preventing the cleavage of the ester bond at the RCL’s P1 residue. Consequently, a stable, covalent serpin–protease complex forms, whereby the bound protease is critically distorted, leaving it inactive and trapped. The conformational change from the native, stressed (S) to the cleaved, relaxed (R) state is called the S to R transition.

The most abundant serpin in circulation is alpha-1-antitrypsin (AAT), with a normal plasma concentration of 100–200 mg/dL that can increase 3–5-fold with systemic infection or inflammation, underscoring its important homeostatic function [6,7]. AAT is synthesized mainly by hepatocytes [8] but also in lesser amounts by monocytes, macrophages, the pancreas, lung alveolar cells, enterocytes, and endothelial cells [9]. The best-known function of AAT is the inhibition of neutrophil elastase, protecting the lungs from an elastase-induced lung injury such as emphysema and bronchiectasis [10]. Neutrophil elastase has both a host-protective effect (against microbial pathogens) and also host-damaging properties, with a significant deficiency in the level and/or function of AAT resulting in the degradation of the lung tissue, culminating in emphysema and/or bronchiectasis. Neutrophil elastase is also activated by an upstream enzyme dipeptidyl peptidase I, for which an inhibitor has entered clinical trials to counteract the damaging effects of excessive elastase [11].

The unusual conformational change required for serpin function provides an elegant opportunity for allosteric regulation by the binding of cofactors, of which the most well-studied is heparin. While heparin is best-known for its anticoagulant properties, it also has other biological activities that include anti-inflammatory, antimicrobial, and anticancer effects [12,13,14,15,16]. The goal of this review is to discuss some of the clinically relevant serine protease–serpin interactions that may be enhanced by heparin or other negatively charged polysaccharides (Table 1) [17,18].

## 2. A Primer on Heparin Biology

All heparins are sulfated (anionic) polysaccharides (glycosaminoglycans) and are considered to have the highest negative charge density of any biologic molecules. “Heparin” is not a distinct molecule with a specific molecular weight but rather comprises varying lengths of sulfated glycosaminoglycans with an estimated molecular weight range of 3000 to 30,000 Daltons (grams/mole). Its major anticoagulant effect is mediated by inactivating thrombin and activated factor X (factor Xa) by an antithrombin III (AT-III)-dependent mechanism. Medical-grade heparins comprise two main formulations: unfractionated heparin (UFH) and low molecular weight heparin (LMWH) such as enoxaparin and nadroparin. Both formulations comprise a mixture of different chain lengths of sulfated glycosaminoglycans. For example, the average molecular weight of UFH is 15,000 Daltons and the range is wide—estimated to be 3000 to ≥30,000 Daltons [23]. In contrast, LMWH is isolated from UFH by various methods of fractionation or depolymerization [24], and although the average molecular weight of enoxaparin and nadroparin are much lower—estimated to be 4300 and 4500 Daltons—even the LMWH comprises glycosaminoglycan chains of varying lengths. The important biological properties of heparin are summarized in Table 2.

## 3. Thrombin, Antithrombin III, and Heparin

Antithrombin III (AT-III) is a serpin that is produced in the liver and that functions as an endogenous anticoagulant by binding to and neutralizing the serine protease activity of the procoagulants factor II (thrombin), factor IXa, and factor Xa. Since the thrombin conversion of fibrinogen to fibrin, along with platelet aggregation, is one of the key steps of blood coagulation, AT-III helps prevent clot formation [28,29,30].

When oligosaccharides are bound to all four glycosylation sites of AT-III, it is known as α-AT-III; when three of the four sites are bound, it is known as β-AT-III [31,32]. Even though β-AT-III has greater intrinsic anticoagulant activity and a greater affinity for endothelial glycosaminoglycans and heparin than α-AT-III, they are both relatively slow inhibitors of thrombin [33]. AT-III, like several other serpins, circulates in plasma in a native but relatively inactive state, where the RCL is partially inserted into the A sheet, positioning the P1 arginine of the RCL away from a protease [19]. Heparin binding to several arginine and lysine residues of AT-III induces a conformational change in the AT-III, expelling the RCL, exposing the P1 arginine, and thus increasing the heparin’s affinity for and inhibition of thrombin [19]. Heparin also acts a bridge by binding to an exosite on the thrombin, further enhancing the association rate of the complex. The heparin binding site in the serpin is formed by the N-terminus, the A-helix, and the D-helix, including their extensions. Since the facilitation of serpin–serine protease interaction by heparin is best-established with the AT-III–thrombin pair, we have drawn a general cartoon based on this specific interaction (Figure 1) but posit that this modeled paradigm (or a similar one) may also be involved with heparin and other serpin–serine protease pairs. X-ray crystallography reveals that, when AT-III binds to heparin, it undergoes conformational changes that bring these regions together, creating a complementary binding site. This arrangement allows AT-III’s basic residues to engage in optimal hydrogen bonding and ionic interactions with heparin’s negatively charged groups. A more comprehensive review of heparin–serpin interactions has been previously published [34]. The representative structures of heparin-binding serpins are shown in Figure 2. Heparin acts as a catalyst since it is released upon formation of the AT-III–thrombin complex [19]. Whereas heparin must bind to both thrombin and AT-III for this facilitated thrombin inhibition, the heparin inhibition of the pro-coagulant activated factor Xa also occurs by the conformational activation of AT-III but does not require the binding of heparin to factor Xa.

The urinary loss of AT-III that occurs with nephrotic syndrome has long been implicated as a cause of the increased risk of venous and arterial thromboembolism, although a more recent study calls into question the importance of this pathogenic mechanism [35]. Since there is an increased risk for thromboembolism in diabetics [36], we speculate that this may be related to increased glycosylation of AT-III, resulting in more of the glycosylated α-AT-III isomer, which has less anticoagulant activity than the less glycosylated β-AT-III.

## 4. Plasmin, Anti-Plasmin, and Heparin

The fibrinolytic process occurs when plasminogen is cleaved by the tissue plasminogen activator (TPA, a serine protease) to plasmin (also a serine protease), which degrades the fibrin-to-fibrin degradation products. Although heparin is clearly an anticoagulant, it also mildly reduces plasmin generation [20] and theoretically may reduce the fibrinolysis of the existing thrombus. Both the TPA and plasmin may also be inhibited by the endogenous serpins—plasminogen activator inhibitor I (PAI-1) and α_2_-anti-plasmin. If the paradigm holds true that heparin augments the serine protease–serpin interaction, it is theoretically possible that heparin may enhance TPA-PAI-1 and plasmin–α_2_anti-plasmin interactions, resulting in the decreased production and activity of plasmin, respectively. We opine that this potential and seemingly paradoxical pro-thrombus property of heparin may be more of a theoretical concern because the ability of heparin to enhance AT-III activity may outweigh its ability to enhance anti-fibrinolytic activity.

## 5. C1 Esterase/Kallikrein, C1-INH, and Heparin

Hereditary angioedema (HAE) is an autosomal dominant disorder (mostly hereditary, although 25% of cases are sporadic), which is caused by the deficiency or dysfunction of the serpin C1 esterase inhibitor (C1-INH, gene *SERPING1*)—known as Type 1 HAE (~85% of cases) and Type 2 HAE, respectively. Since C1-INH is a promiscuous serpin that is capable of inhibiting other serine proteases, a lack of functional C1-INH results in the decreased inhibition of kallikrein and plasmin, which are serine proteases that convert high molecular weight kininogen to bradykinin, a downstream vasoactive mediator of HAE [37]. Hence, a consequence of decreased C1-INH activity is an unopposed increase in kallikrein and plasmin activity, resulting in increased bradykinin production. Upon the binding of excess bradykinin to its B2 receptor on endothelial cells, vasodilation and increased permeability of the post-capillary venules occur, culminating in the capillary leak and edema that characterize HAE.

The current pharmacologic therapies for acute attacks of HAE include C1-INH replacement (plasma-derived or recombinant), ecallatide (a kallikrein inhibitor), and icatibant (a bradykinin receptor antagonist) [37,38,39,40]. These costly but effective agents are rarely associated with serious side effects, although there is the risk of anaphylaxis with C1-INH replacements and ecallatide. Long-term prophylactic agents to lower the attack rate of acute HAE or to shorten the episodes include plasma-derived C1-INH, a monoclonal antibody targeting kallikrein (lanadelumab), and small molecular weight kallikrein inhibitors (berotralstat and avoralstat) given orally [41,42,43,44].

Nearly 100 years ago, heparin was reported to potentiate C1-INH activity in the classical complement pathway [45,46]. Hypothesized mechanisms include the ability of heparin to enhance the binding of C1-INH to C1 esterase (C1s) and to interfere in the binding of C1q to an activator (Figure 2) [46,47,48]. While the complement pathway per se is not considered to play a pathogenic role in HAE, as previously mentioned, C1-INH also inhibits kallikrein and plasmin, the serine proteases that produce bradykinin. Thus, the ability of the heparins to augment C1-INH activity may be highly relevant in the context of preventing or treating acute attacks of HAE. Indeed, UFH and LMWH have shown efficacy in the treatment of acute HAE attacks in case reports and series [21,49,50,51].

One likely mechanism of how negatively charged polysaccharides may augment the interaction of serine protease with its serpin is through inducing a more favorable interaction between the pair. For example, it is well known that the polysaccharide dextran can potentiate C1-INH activity [17]. This facilitation is supported by the finding that one negatively charged dextran molecule binds to multiple positively charged F1 helices that are present in C1-INH, neutralizing some of the relevant positive charges on lysine and arginine residues on the C1-INH. This interaction facilitates the binding of C1-INH to the positively charged autolysis loop of serine proteases, forming a ternary complex of polysaccharide–C1-INH–protease and the polysaccharide that is not “sandwiched” between the protease and anti-protease [17].

## 6. Furin, TMPRSS2, and Their Inhibition by AAT in the Context of HIV, the Hepatitis C Virus, and/or SARS-CoV-2

Specific envelope proteins of several viruses need to be processed prior to fusion of their viral envelopes with cell membranes. This cleavage of viral envelope proteins is performed in several instances by cell surface serine proteases. With the HIV-1 infection, furin, a cell surface serine protease, cleaves gp160, a key glycoprotein of the HIV-1 envelope, to gp120 and gp41. Located on the viral surface, gp120 mediates the viral attachment to the cell surface by binding to the CD4 receptor and co-receptors CCR5 and CXCR4. Following the attachment, gp41, a transmembrane protein, facilitates the fusion of the viral and cell membranes. Three decades ago, an engineered form of AAT known as AAT Portland (α_1_-PDX) was found to inhibit furin; thus, by inhibiting the formation of gp120, this synthetic AAT prevented the fusion of the HIV-1 membrane to cell membranes [52,53,54,55]. Shapiro and co-workers [56] also demonstrated that AAT inhibited HIV-1 replication in various cell cultures; furthermore, they showed that, whereas HIV-1 did not replicate in the blood of healthy volunteers, robust replication occurred in the blood of AAT-deficient subjects. While heparin enhances the furin cleavage of HIV-1 and gp160 [57], which facilitates an HIV-1 infection, whether heparin would augment the AAT inhibition of furin, to the best of our knowledge, has not been examined. 

Five years before the COVID-19 pandemic, AAT was reported to inhibit TMPRSS2 (serine protease Transmembrane Protease 2), a cell surface serine protease that is required to activate the hepatitis C virus (HCV) [58]. Furthermore, among patients with a known AAT deficiency, the presence and absence of an HCV infection displayed a very good positive and negative predictor for the presence of chronic liver disease [59]. Since we have found that enoxaparin can enhance the AAT inhibition of TMPRSS2 [18], it seems plausible that heparin may also augment the AAT inhibition of an HCV infection.

Following the binding of SARS-CoV-2, the causative agent of COVID-19, via its spike protein to the cell surface receptor ACE2 (angiotensin converting enzyme 2), both furin and TMPRSS2 sequentially cleave the spike protein at the S1/S2 and S2’ sites, respectively. This processing of the spike protein by host serine proteases is required for the coronavirus envelope to fuse with the cell membrane prior to viral entry into the cells [60,61].

The serpin HAI-2 is a known endogenous inhibitor of TMPRSS-2 [62]. Two pharmacologic inhibitors of TMPRSS2 include camostat and nafamostat. While camostat is a proven inhibitor of TMPRSS2 and has been shown in vitro to inhibit a SARS-CoV-2 infection [61,63], a randomized, double-blind, placebo-controlled trial of ~200 hospitalized COVID-19 patients showed that camostat had no effect on clinical worsening or mortality [64].

In both human cell lines and primary human airway epithelial cells (hAEcs), AAT was demonstrated to inhibit TMPRSS2 activity, decrease a SARS-CoV-2 infection, or to achieve both [18,65,66,67]. Oguntuyo and co-workers [68] showed that sera from SARS-CoV-2-naïve individuals inhibited the cellular entry of SARS-CoV-2 and identified AAT as the inhibitory molecule responsible. Elastase, the canonical serine protease strongly inhibited by AAT, also cleaves the spike protein at the S1/S2 site of a SARS-CoV-2 variant (D614G) [69]. A three-dimensional lung cell model using primary cells from multiple donors found that more resistant donors to an in vitro SARS-CoV-2 infection expressed higher transcript levels of *SERPINA1* (AAT gene), *SERPINE1*, and *SERPINE2* in the airway cells [70]. Variants of SerpinB3—through mutagenesis of its RCL—have been generated and found to be more potent in inhibiting TMPRSS2 than AAT [71]. We found that the AAT inhibition of TMPRSS2 activity in primary hAEcs was augmented significantly by enoxaparin [18]. Detailed in silico modeling demonstrated the following: *(i)* there is a suboptimal electrostatic complementarity between AAT and TMPRSS2 (adjacent electropositive blue patches) at both the buried interface and the solvent-exposed interface rim in a molecular charge model (Figure 3A) that is stabilized by bridging negatively charged heparin molecules (Figure 3A, multi-colored stick figures); *(ii)* the RCL of AAT (Figure 3B, magenta) adopts an inhibitory-competent conformation compared with the crystal structure of TMPRSS2 (Figure 3B, cyan); superimposed is the bound exogenous inhibitor of TMPRSS2 (nafamostat) (Figure 3B, wheat) or the endogenous inhibitor of TMPRSS2 (HAI-2) (Figure 3C, wheat); and *(iii)* negatively charged heparin (Figure 3D, gray stick figures) bridges the adjacent electropositive amino acid residues (K = lysine, R = arginine) at the TMPRSS2–AAT interface, neutralizing otherwise repulsive forces [18].

We posit that the combination of AAT and enoxaparin will be more effective against COVID-19 than agents that only inhibit TMPRSS2 such as camostat [64], because AAT and enoxaparin individually and synergistically antagonize the SARS-CoV-2 infection and several other pathogenic mechanisms of COVID-19 (e.g., AAT is anti-inflammatory, inhibits neutrophil elastase, and protects against endothelial cell death, whereas heparin inhibits coagulation, is anti-inflammatory, competitively inhibits SARS-CoV-2 from binding to heparan sulfate (a co-receptor for SARS-CoV-2), and augments the C1-INH inhibition of kallikrein, preventing both bradykinin formation and capillary leak [72]).

These laboratory investigations are supported by epidemiologic and clinical evidence—either in groups known to be AAT-deficient or in non-select groups—that absolute or relative AAT deficiency increases the risk of COVID-19 infection, more severe disease, and/or greater mortality [73,74,75,76,77,78,79,80,81,82,83,84,85,86]. Furthermore, the evidence implicates AAT deficiency itself—and not its pulmonary sequelae of emphysema or bronchiectasis—as a major risk factor for the SARS-CoV-2 infection [78]. Even without an AAT deficiency, the AAT response may be insufficient in critically ill COVID-19 patients [87,88], perhaps due to the AAT heterozygosity (e.g., the protease inhibitor (Pi)MZ) that results in a suboptimal increase in AAT levels [6] or to the oxidation of normal AAT in highly oxidative states such as severe COVID-19, causing loss of its serpin activity [89] or inducing its polymerization [9,90]. Interestingly, the IL-6 receptor antagonists (e.g., tocilizumab) that are used as an adjunctive anti-inflammatory treatment for patients with severe COVID-19 recalcitrant to glucocorticoids may reduce AAT expression (since IL-6 induces AAT) [91].

## 7. TMPRSS2, Prostate Cancer, and Heparin

While those with AAT deficiency due to the PiZZ genotype are at risk of liver cancer, this is likely due to the accumulation of Z-AAT polymers causing liver inflammation and cirrhosis that progressed to cancer, i.e., a toxic gain of function. Neutrophil-associated inflammation may also help drive tumorigenesis that may be inhibited by AAT [92]. Interestingly, the gene that encodes *TMPRSS2* is also androgen-responsive and a *TMPRSS2* gene fusion with ETS transcription factors (such as v-ets erythroblastosis virus E26 oncogene homolog) results in the *TMPRSS2:ERG* gene product that is increasingly recognized to be involved in prostate carcinogenesis [93,94]. There is also increasing evidence that heparin products may antagonize cancer pathogenesis through the inhibition of proliferation, adhesion, angiogenesis, migration, and invasion of cancer cells [15]. Thus, it is interesting to speculate that the inhibition of TMPRSS2 by AAT plus heparin may play a role in antagonizing the pathogenesis and progression of certain forms of cancer.

## 8. Future Directions

The paradigm that heparin augments the inhibitory activity of serpin on serine proteases may be more widespread than presently known. While there may be a specific mechanism for each specific serine protease–serpin pair, induction by the negatively charged heparin molecule of a more favorable electrostatic interaction may be a fundamental underlying mechanism. Other potential effects of heparin on serine protease–serpin interactions remain unexplored. For example, TMPRSS2 also facilitates infection of the respiratory epithelium by the influenza A virus, and AAT may inhibit this effect, although the mechanism remains unknown [95]. Additionally, as shown in Table 3, some of the serpin discussed herein are known to inhibit serine proteases other than their cognate serine protease and thus could be addressed in future studies in the context of heparin. Another unknown area is whether heparin could facilitate the AAT antagonism of neutrophil extracellular traps—which have both antibacterial and injurious inflammatory properties, as seen with bronchiectasis and emphysema—in which neutrophil elastase is a component. While heparin products have traditionally been administered intravenously or subcutaneously, the delivery of inhaled heparin may be particularly useful for lung-related conditions such as acute lung injury, while potentially reducing the systemic side effects that may be seen with parenteral administration [96].

## Figures and Tables

**Figure 1 ijms-25-01804-f001:**
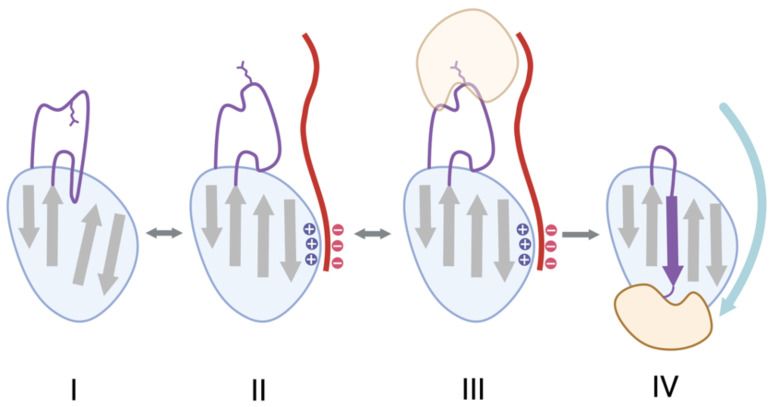
Serpin activation by heparin as exemplified by the thrombin–AT-III–heparin interaction. (**I**) The RCL (purple) of the serpin AT-III is partially inserted into the A β-sheet (the grey β-strands), positioning the P1 arginine of the RCL (the purple sticks) away from the thrombin protease. (**II**) Negatively charged heparin (in red) binds to positively charged patches of AT-III, inducing a conformational change in the AT-III, expelling the RCL, and exposing the P1 arginine, thus increasing the heparin’s affinity for the thrombin (in brown). (**III**) Heparin acts a bridge by binding to an exosite on the thrombin, further enhancing the association rate of the complex. (**IV**) The RCL integrates between strands 3 and 5 of the A β-sheet, translocating the protease to the other end of the serpin, resulting in a stable, covalent serpin–protease complex and in the release of heparin.

**Figure 2 ijms-25-01804-f002:**
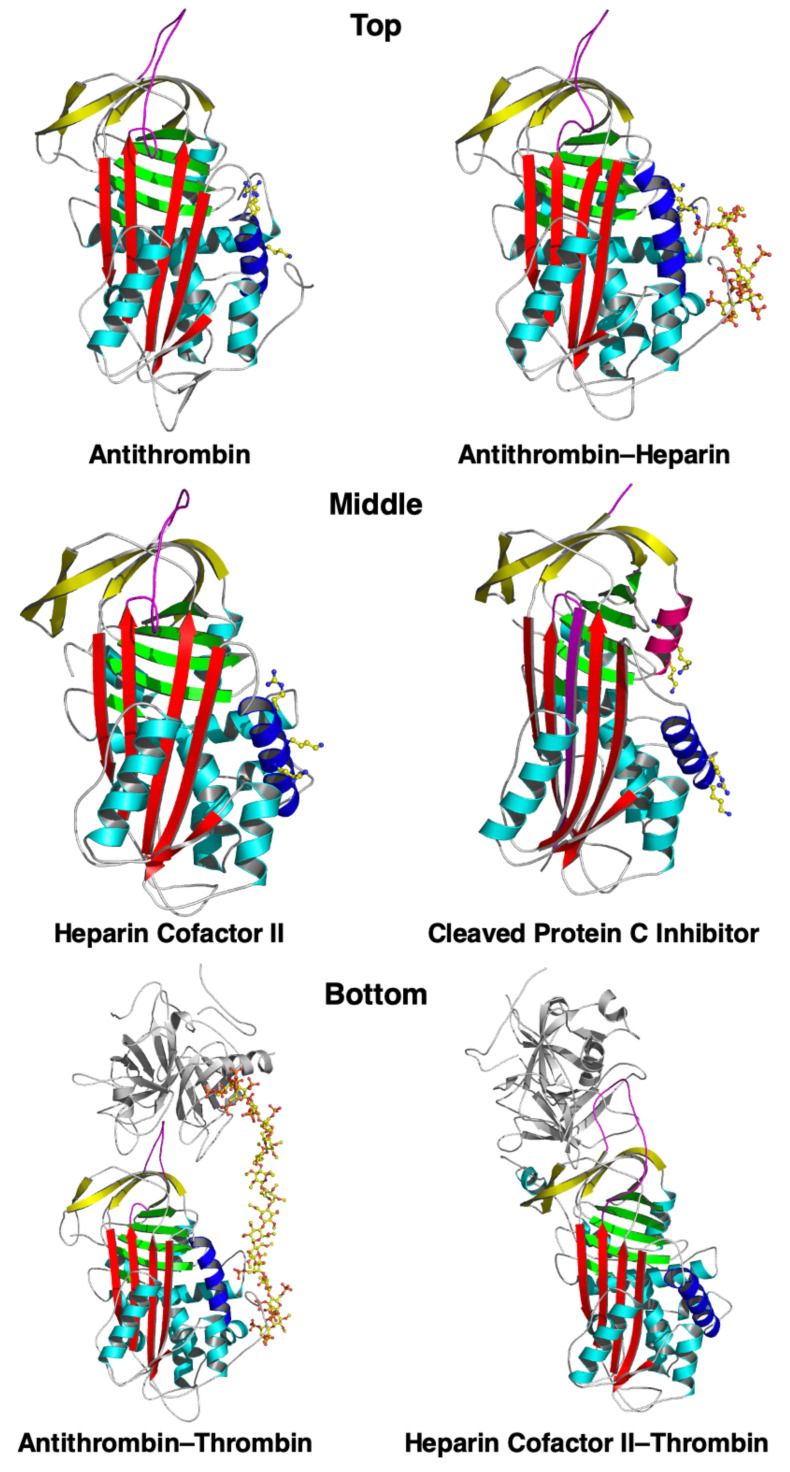
Structures of three heparin-binding serpins. (**Top**) The structures of antithrombin without ((**left**) panel; PDB ID 1ANT) and with ((**right**) panel; PDB ID1E03) the heparin pentasaccharide bound. (**Middle**) Heparin cofactor II ((**left**) panel; PDB ID 1JMJ) and cleaved protein C inhibitor ((**right**) panel; PDB ID 1LQ8) are shown as indicated. In all of the structures, the A β-sheet is shown in red, the B β-sheet in green, and the C β-sheet in yellow. The reactive center loop is shown in magenta and the D-helix in blue. The H-helix is shown in dark pink in the protein C inhibitor structure. Positively charged residues on the D- and H-helices and the heparin pentasaccharide are shown in a ball and stick format. (**Bottom**) The bimolecular encounter of complex structures of antithrombin–thrombin ((**left**) panel; PDB ID 1TB6) and of heparin cofactor II–thrombin (S195A active site mutant) ((**right**) panel; PDB ID 1JMO), using the same color scheme as shown above.

**Figure 3 ijms-25-01804-f003:**
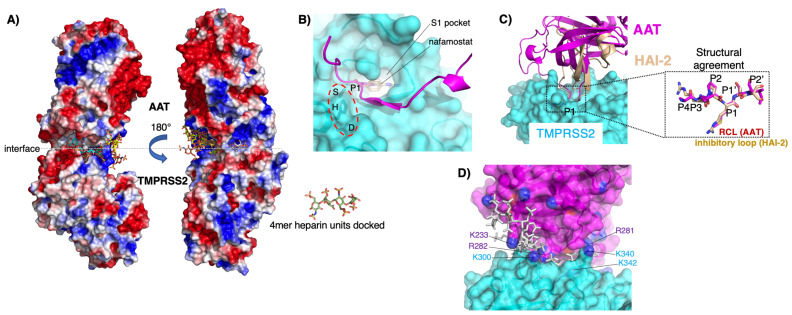
In silico modeling demonstrating AAT binding to TMPRSS2 and how this interaction may be augmented by heparin. (**A**) Heparin molecules that stabilize the TMPRSS2–AAT association by acting as electrostatic bridges; electrostatic potential surfaces of the TMPRSS2–AAT complex (blue = positive, red = negative) showing docked heparin 4 mers binding to electropositive patches on the molecular surface. The carbon atoms of each 4 mer heparin are colored differently. (**B**) Molecular surface of TMPRSS2 showing bound RCL (magenta cartoon). The catalytic residues are labelled (with a red dashed oval). The P1 sidechain of the AAT occupies the S1 site of the TMPRSS2 with superimposition of the TMPRSS2 inhibitor nafamostat (wheat). (**C**) Structural agreement between the RCL of AAT and the inhibitory loop of HAI-2; model of the complex between TMPRSS2 and an endogenous TMPRSS2 inhibitor, HAI-2, structurally aligned with the TMPRSS2–AAT complex. The TMPRSS2 is shown as a transparent molecular surface (cyan); the AAT (magenta) and HAI-2 (wheat) as cartoons. The inset shows a closeup comparison of the inhibitory loops of HAI-2 (wheat) and AAT–RCL (magenta) that interact with the TMPRSS2. (**D**) Location of lysine (K) and arginine (R) residues (AAT = magenta; TMPRSS2 = cyan) at the TMPRSS2 (cyan)–AAT (magenta) interface. A semi-transparent molecular surface is shown, and the lysine and arginine residues that contribute to unfavorable electrostatics at the interface are labelled (the sidechain nitrogen atoms are colored blue). Negatively charged heparin molecules (grey sticks) are shown, stabilizing the otherwise unfavorable interactions between the positively charged lysine and arginine residues between the TMPRSS2 and AAT.

**Table 1 ijms-25-01804-t001:** Serine protease–serpin interactions augmented by negatively charged polysaccharides.

Serine Protease and Serpin Pair	Augmentation by Negatively Charged Polysaccharide
Thrombin and antithrombin III	Heparin [19]
Plasmin and anti-plasmin	Heparin [20]
C1-esterase and C1-INH	Heparin (nadroparin and enoxaparin)Dextran [17]
C1-esterase and kallikrein	Heparin, nadroparin [21]
Furin and AAT Portland	Unknown
TMPRSS2 and AAT	Enoxaparin, UFH [18]
Neutrophil elastase and AAT	Heparin [22]
Trypsin and AAT	Heparin [22]

**Table 2 ijms-25-01804-t002:** Protean biological activities of heparin [12,13,14,15,16].

Biologic Effect	Example(s)
Anticoagulation	Inhibits coagulation through inactivation of thrombin and factor X.
Antiviral	Competitively inhibits cell surface heparan sulfate, a co-receptor SARS-CoV-2.
Antibacterial	Possibly inhibits growth of certain bacteria, including *Staphylococcus aureus* [25].
Antimycobacterial	Reduces hepcidin expression (an iron-sequestering protein) in macrophages infected with *Mycobacterium tuberculosis* (*MTB*), increases ferroportin expression (an iron exporter protein), and decreases the intracellular burden of *MTB* [26].
Anti-elastase and antitrypsin	Inhibits the enzymatic activity of neutrophil elastase on its release from neutrophils [27] and enhances the inhibitory effect of AAT on elastase and trypsin [22].
Anti-inflammatory	Inhibits inflammation by inhibiting nuclear factor-kappa B (NFκB) activation, binding to pro-inflammatory molecules (IL-8, major basic protein, and complement components), and reducing the release and activity of IL-6.
Anticomplement and contact system	Augments C1-esterase inhibitor (C1-INH) inhibition of kallikrein ⟶ a decrease in capillary leak.
Cell protection	Binds to extracellular histones released from dead cells, mitigating histone-mediated endothelial and organ dysfunction. Interacts with endothelial cells to maintain vascular integrity.
Anticancer	Blocks angiogenesis and adhesion of cancer cells to platelets, potentially interrupting the metastases of cancer cells.

**Table 3 ijms-25-01804-t003:** Serine proteases known to be inhibited by the serpins discussed.

Serpin	Serine Proteases Inhibited
Antithrombin III	Thrombin, factor Xa, kallikrein, and to a lesser extent IXa and XIIa [97,98].
Anti-plasmin	Plasmin and plasminogen.
C1-esterase inhibitor	C1 esterase (C1r, C1s), MASP2, plasmin, tissue plasminogen activator, factor XI, factor XIIa, and kallikrein [99].
Alpha-1-antitrypsin	Proteinase-3, trypsin, chymotrypsin, myeloperoxidase, cathepsins, a-defensins, tryptase, plasmin, thrombin, factor Xa, urokinase, a disintegrin and metalloprotease 17 (ADAM17, aka tumor necrosis factor converting enzyme), and Transmembrane Protease 2 (TMPRSS2) [18,72,100,101].

## Data Availability

Not applicable.

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
