# Peer review of "The Inhibition of Serine Proteases by Serpins Is Augmented by Negatively Charged Heparin: A Concise Review of Some Clinically Relevant Interactions"

_ijms, 2024, doi:10.3390/ijms25031804_

Round 1
Reviewer 1 Report
Comments and Suggestions for Authors
The manuscript ijms-2811864 reviews the impact of heparin on the inhibition of serine protease by serpin. The manuscript is well written, and I recommend the publication after minor revisions as follows below:
1) Provide at the beginning of the manuscript a Scheme and explanation of the mechanisms of serine protease in terms of nucleophilic, electrophilic, and the necessity or not of hydration molecules in the catalysis.
2) Are the inhibitors explored in this manuscript considered pan-serine protease inhibitors? Please, make a critical review about it.
3) Please, provide as a Table the type of inhibition mechanism induced by the inhibitors provided in this manuscript, e.g., allosteric, competitive, non-competitive…
Comments on the Quality of English Language....
Author Response
REVIEWER 1
The manuscript ijms-2811864 reviews the impact of heparin on the inhibition of serine protease by serpin. The manuscript is well written, and I recommend the publication after minor revisions as follows below:
Comment 1: Provide at the beginning of the manuscript a Scheme and explanation of the mechanisms of serine protease in terms of nucleophilic, electrophilic, and the necessity or not of hydration molecules in the catalysis.
Response: We like this suggestion and have included a detailed discussion of the nucleophilic and electrophilic mechanisms by which the serine residue of the serine protease cleaves a peptide bond. This new text is located in the first paragraph of the Introduction, page 3 of the tracked version.
Comment 2: Are the inhibitors explored in this manuscript considered pan-serine protease inhibitors? Please, make a critical review about it.
Response: Anti-plasmin is relatively specific serine protease inhibitors for plasmin and plasminogen. However, the other serpins are more promiscuous and are able to inhibit several serine proteases. We have included a new Table 3 of the serine proteases known to be inhibited by the serpins discussed in this review. These interactions could be studied in the future in the context of heparin. The relevant text and new Table 3 are in Section VIII (Future directions).
Comment 3: Please, provide as a Table the type of inhibition mechanism induced by the inhibitors provided in this manuscript, e.g., allosteric, competitive, non-competitive…
Response: All serpins act as irreversible, suicide inhibitors, in contrast to the competitive, or ‘lock-and-key’ inhibitors such as Kunitz-type inhibitors, as mentioned already on page 2 (starting at line 62 of the CLEAN revision). Thus, a discussion of whether the inhibitory of mechanism is allosteric, competitive, or non-competitive would not be relevant here. Their function can be regulated by the allosteric interactions with other molecules, such as heparin – the topic of this review. We have included a new Table 3, however, to show the serpins discussed and the proteases they inhibit.
Reviewer 2 Report
Comments and Suggestions for Authors
Journal: International Journal of Molecular Sciences
Manuscript ID: ijms-2811864
Type of manuscript: Review
Title: Inhibition of serine proteases by serpin is augmented by negatively charged heparin: A concise review of some clinically relevant interactions
Authors: Edward D. Chan *, Paul T. King, Xiyuan Bai, Robert A. Sandhaus, Ashley Buckle
Judgment of this reviewer is “minor revision”.
In this review, the authors briefly summarize pairs of serine proteases and their inhibitory proteins (SERPINs), which are specifically relevant to medicine. By adding an explanation of the role of heparin as a cofactor, it adds a new perspective to its medicinal efficacy.
Please correct the following formal errors.
L005: The instruction (https://www.mdpi.com/journal/ijms/instructions#preparation) says
“Please do not include abbreviated or short forms of the title, such as a running title or head. These will be removed by our Editorial Office.”. Is a running title necessary? This reviewer has looked at the instructions and past papers, but can't find it.
L007: The position of 'and' is wrong.
L050-72: Since this is the core of the content that follows in the paper, it is a good idea to include an illustration of an organized conceptual diagram.
Appropriate citations are the core of a review article.
L107-108: References for the fact of LMWH as UFH.
L132-133: References for the fact of the conformational change upon heparin binding.
L136: for Figure 1, if the authors use the information of the Protein Data Bank, the PDB IDs are to be indicated for the convenience.
L304, 311: References or explanations for “PiMZ” and ”PiZZ”.
L315, 317: wrong fonts
Author Response
REVIEWER 2
In this review, the authors briefly summarize pairs of serine proteases and their inhibitory proteins (SERPINs), which are specifically relevant to medicine. By adding an explanation of the role of heparin as a cofactor, it adds a new perspective to its medicinal efficacy.
Comment 1: L005: The instruction (https://www.mdpi.com/journal/ijms/instructions#preparation) says
“Please do not include abbreviated or short forms of the title, such as a running title or head. These will be removed by our Editorial Office.”. Is a running title necessary? This reviewer has looked at the instructions and past papers, but can't find it.
Response: Thank you for clarifying. The running title has been removed from the title page.
Comment 2: L007: The position of 'and' is wrong.
Response: This has been corrected.
Comment 3: L050-72: Since this is the core of the content that follows in the paper, it is a good idea to include an illustration of an organized conceptual diagram.
Response: We have added a cartoon that shows conceptually how we posit heparin may be augmenting the serpin-serine protease interaction (new Figure 1). In the legend of this Figure 1, we have described the Figure in the context of the anti-thrombin III and thrombin interaction since this mechanism is very well characterized.
Comment 4: L107-108: References for the fact of LMWH as UFH.
Response: A reference has been added for the derivation of LMWH from UFH in Section II (A primer on heparin biology).
Comment 5: L132-133: References for the fact of the conformational change upon heparin binding.
Response: The reference we had used for this text is correct as it discusses conformation change of AT-III upon binding of heparin (Ersdal-Badju E et al. J Biol Chem 1997).
Comment 6: L136: for Figure 1, if the authors use the information of the Protein Data Bank, the PDB IDs are to be indicated for the convenience.
Response: We have added the PBD IDs to relabeled Figure 2 legend.
Comment 7: L304, 311: References or explanations for “PiMZ” and ”PiZZ”.
Response: Pi has now been defined as “protease inhibitor.”
Comment 8: L315, 317: wrong fonts
Response: These have been corrected.